# Multiple Antimicrobial Effects of Hybrid Peptides Synthesized Based on the Sequence of Ribosomal S1 Protein from *Staphylococcus aureus*

**DOI:** 10.3390/ijms23010524

**Published:** 2022-01-04

**Authors:** Sergey V. Kravchenko, Pavel A. Domnin, Sergei Y. Grishin, Alexander V. Panfilov, Viacheslav N. Azev, Leila G. Mustaeva, Elena Y. Gorbunova, Margarita I. Kobyakova, Alexey K. Surin, Anna V. Glyakina, Roman S. Fadeev, Svetlana A. Ermolaeva, Oxana V. Galzitskaya

**Affiliations:** 1Institute of Environmental and Agricultural Biology (X-BIO), Tyumen State University, 625003 Tyumen, Russia; svkraft@yandex.ru; 2Biology Faculty, Lomonosov Moscow State University, 119991 Moscow, Russia; paveldomnin6@gmail.com; 3Gamaleya Research Centre of Epidemiology and Microbiology, 123098 Moscow, Russia; drermolaeva@mail.ru; 4Institute of Protein Research, Russian Academy of Sciences, 142290 Pushchino, Russia; syugrishin@gmail.com (S.Y.G.); panfilov.alexander@mail.ru (A.V.P.); alan@vega.protres.ru (A.K.S.); quark777a@gmail.com (A.V.G.); 5The Branch of the Institute of Bioorganic Chemistry, Russian Academy of Sciences, 142290 Pushchino, Russia; viatcheslav.azev@bibch.ru (V.N.A.); mustaeva@rambler.ru (L.G.M.); eyugorbunova@rambler.ru (E.Y.G.); 6Institute of Theoretical and Experimental Biophysics, Russian Academy of Sciences, 142290 Pushchino, Russia; kobyakovami@gmail.com (M.I.K.); fadeevrs@gmail.com (R.S.F.); 7State Research Center for Applied Microbiology and Biotechnology, 142279 Obolensk, Russia; 8Institute of Mathematical Problems of Biology and Keldysh Institute of Applied Mathematics, Russian Academy of Sciences, 142290 Pushchino, Russia

**Keywords:** ribosomal S1 protein, amyloid, antimicrobial peptides, *Staphylococcus aureus*, MRSA, cell-penetrating peptide, FoldAmyloid, AlphaFold 2

## Abstract

The need to develop new antimicrobial peptides is due to the high resistance of pathogenic bacteria to traditional antibiotics now and in the future. The creation of synthetic peptide constructs is a common and successful approach to the development of new antimicrobial peptides. In this work, we use a simple, flexible, and scalable technique to create hybrid antimicrobial peptides containing amyloidogenic regions of the ribosomal S1 protein from *Staphylococcus aureus*. While the cell-penetrating peptide allows the peptide to enter the bacterial cell, the amyloidogenic site provides an antimicrobial effect by coaggregating with functional bacterial proteins. We have demonstrated the antimicrobial effects of the R23F, R23DI, and R23EI hybrid peptides against *Staphylococcus aureus*, methicillin-resistant *S. aureus* (MRSA), *Pseudomonas aeruginosa*, *Escherichia coli*, and *Bacillus cereus*. R23F, R23DI, and R23EI can be used as antimicrobial peptides against Gram-positive and Gram-negative bacteria resistant to traditional antibiotics.

## 1. Introduction

In recent years, interest in the study and development of antimicrobial peptides (AMPs) has been growing, which suggests a stable long-term trend [1,2,3]. This interest is mainly related to the urgent need to combat the spread of multidrug-resistant pathogens to classical antibiotics, as well as progress in the development of tools that predict the antimicrobial properties of peptides [4,5]. The advantage of AMPs over traditional antibiotics lies in their multiple spectra of action, and the rapid antimicrobial action of AMPs in combination with biodegradability; as a rule, low toxicity can be supplemented by the use of composites, various expression, and delivery systems [6,7,8]. However, although that some successful examples of the effective use of synthetic peptides to fight pathogenic bacteria have been shown, there is no guarantee that the final sequence will be effective against a specific pathogen, stable, and non-toxic to human cells [9]. From the point of view of solving the problem of developing new antimicrobial drugs, interdisciplinary approaches can be used that combine the computational capabilities of predicting several structural and physiological properties of active molecules. At the same time, it is necessary to develop peptides that, under nearly physiological conditions in vitro (neutral pH, medium ionic strength, temperature 37 °C), form ordered structures, since the ability to self-assemble is important for the manifestation of action of antimicrobial substances [10]. These properties are possessed by amyloidogenic peptides, some of which have been described as efficient antimicrobial agents [11,12,13].

The amyloidogenicity of peptides and proteins can be assessed using various prediction algorithms such as FoldAmyloid [14], AGGRESCAN [15], Pasta 2.0 [16], and Waltz [17]. These and similar bioinformatics tools allow to identify protein aggregation-prone regions/potential amyloidogenic determinants (PADs) [18,19]. PADs can be predicted using a variety of physical and statistical models, which have recently been increasingly used in combination with neural networks and machine learning algorithms [20,21]. At the same time, the classification and labeling of the training dataset have a decisive influence on the prediction results for such models. It is assumed that the accuracy of predicted amyloidogenicity can be improved by annotating peptides not only as amyloid/non-amyloid but also as “prone to form oligomers” [22]. The formation of oligomers is an important stage on the path of amyloid assembly [23,24]. In addition, oligomerization is used by antimicrobial peptides as part of their mechanism of action, for example, for the formation of channels and pores in bacterial membranes [25,26].

In this regard, it is of interest to use hybrid peptides that contain fragments of cell-penetrating peptides and protein regions prone to the formation of ordered aggregates [27]. They allow to deliver “cargo” into the cell and do not have significant membranolytic effects [28]; in addition, they perform specific functions of peptides associated with self-assembly [27]. Based on the recent results of studying hybrid peptides based on combinations of TAT-peptide fragments and amyloidogenic regions of bacterial ribosomal S1 proteins, we suggest that amyloidogenic propensity may significantly contribute to the antimicrobial effect of synthetic antimicrobial peptides [29].

We used the ribosomal S1 protein from *S. aureus* as a target protein for the development of amyloidogenic peptides and the creation of hybrid AMPs (Figure 1). The S1 protein is encoded by the rpsA gene, which is part of the S1 operon [30]. An important structural characteristic of the S1 protein is its domain composition [31,32] so that each domain is a conserved OB-fold motif called the S1 domain [33]. It has been demonstrated that N-terminal domains are important in interacting with ribosomal proteins while C-terminal domains interact with mRNA [34]. Identity analysis of S1 domains can be useful for the development of universal peptides for targeted coaggregation with the most conserved amyloidogenic regions of the S1 protein sequence [35]. It is known that the amyloidogenicity of protein domains may have a functional significance for the interaction of a protein with other proteins and nucleic acids [36]. At the same time, the amyloidogenicity of the domains of ribosomal protein S1 plays a key role in its tendency to aggregation [37,38]. It is known that amyloidogenic proteins and peptides of bacteria can participate in biofilm formation and thus enhance bacterial infection [39]. On the other hand, coaggregation of hybrid peptides with amyloidogenic regions of ribosomal proteins of bacteria can lead to disruption of their functioning, which means that such hybrid peptides can exhibit antimicrobial effects [29,40]. The targeted action of the R23I peptide was shown in experiments on *T. thermophilus*. Thus, according to the data of mass spectrometry analysis, the S1 protein was absent in bacterial preparations treated with effective concentrations of the R23I peptide [27].

In this article, we focused on testing the antimicrobial action of peptides against microorganisms that differ in antibiotic resistance, bacterial wall structure, and the number of domains in ribosomal S1 proteins. We predicted the tertiary structure of S1 from *S. aureus*, and also created 3D models of the synthesized hybrid peptides using the AlphaFold 2 algorithms [41]. The antimicrobial effects of the hybrid peptides R23F, R23DI, R23EI have been assessed against strains of pathogenic bacteria *S. aureus*, *B. cereus*, *E. coli*, and *P. aeruginosa* on agar and in liquid media. The effects of inhibition of cell growth in the liquid medium after co-incubation with peptides were compared to data on dead and living cells obtained from fluorescence microscopy images. The possible cytotoxic effect of peptides on eukaryotic cells was investigated using tests for the survival of human fibroblast cells and a line of breast tumor cells. Statistical analysis of the possible cytostatic and antiproliferative effects of the R23F, R23DI, R23EI peptides was carried out for data obtained on eukaryotic cells.

## 2. Results

### 2.1. Prediction of Amyloidogenicity of Ribosomal S1 Protein from S. aureus

We were guided by the idea that a peptide with an amyloidogenic region will interact with a target protein containing the same amyloidogenic site. We sequentially tested the developed antimicrobial peptides containing amyloidogenic regions of the ribosomal S1 protein, first against model organisms (*T. thermophilus* and *E. coli*) [27], then against pathogenic microorganisms (*P. aeruginosa* and *S. aureus*) [29].

The identity of the S1 protein domains from bacteria was investigated earlier [35]. Analysis of the 1453 sequences of ribosomal S1 proteins from 25 different phyla showed that D3 has the highest identity and amyloidogenicity. To exploit the amyloidogenic properties of antimicrobial peptides, amyloidogenic regions of the specific S1 protein from *S. aureus* were selected and candidates for AMPs were developed. The consensus regions predicted by programs were selected as base sequences for the synthesized peptides: VVVHINGGKF (V10F), GVVVRLANFG (G10G), VQGLVHISEI (V10I), and QQVNVKILGI (Q10I) (Figure 2).

Thus, amyloidogenic regions are not predicted in amino acid sequences outside of the S1 domains. Moreover, amyloidogenic regions were predicted in each of the four S1 domains. This is a significant result for the selection of the amyloidogenic sequence of future hybrid peptides and, in general, corresponds to the revealed high amyloidogenicity of S1 domains of S1 proteins of various organisms [38]. Most of the synthesized peptides (three out of four) belong to D4 domain of the ribosomal S1 protein from *S. aureus*.

**Figure 2 ijms-23-00524-f002:**
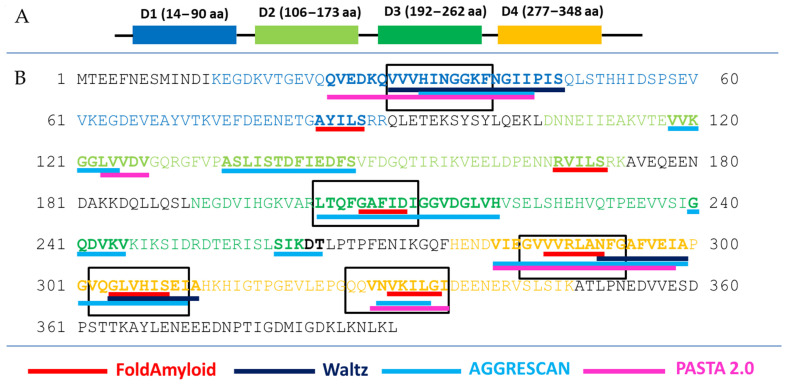
Schematic representation of the S1 domain organization from *S. aureus* (**A**) and the results of prediction of amyloidogenic regions (**B**). Amino acid sequences included in the domains are color-coded—blue (domain 1, D1), light green (domain 2, D2), dark green (domain 3, D3), orange (domain 4, D4). According to the prediction results, amyloidogenic regions are in bold and underlined in red (FoldAmyloid), dark blue (Waltz), light blue (AGGRESCAN), and pink (PASTA 2.0). Peptides synthesized according to the predictions of the FoldAmyloid, Waltz, AGGRESCAN, and PASTA 2.0 programs are highlighted with black rectangles. Predicted amino acid residues are highlighted in bold by at least one program.

### 2.2. Construction of Hybrid Peptides

We performed a preliminary analysis (Appendix A) of the antimicrobial action of peptides VVVHINGGKF (V10F), GVVVRLANFG (G10G), VQGLVHISEI (V10I), QQVNVKILGI (Q10I) against *S. aureus, MRSA*, *B. cereus,* and *E. coli* bacteria. According to the results of most experiments, the peptides did not show antimicrobial action. Nevertheless, the V10F and V10I peptides suppressed the growth of MRSA SA 180-F strain cells in some tests (Appendix A). Based on the results, only two peptides (V10F and V10I) were selected for the development of hybrid peptides. Since the third domain is the most conserved among S1 proteins of bacteria [35], a hybrid peptide based on the sequence LTQFGAFIDI (L10I) from D3 of S1 protein (*S. aureus*) was also synthesized. Thus, three hybrid peptides were synthesized based on the amyloidogenic regions (V10F, L10I, and V10I from D1, D3, and D4 respectively): RKKRRQRRRGGSarGVVVHI-Asi-GGKF-NH_2_ (R23F), RKKRRQRRRGGSarGLTQFGAFIDI-NH_2_ (R23DI), and RKKRRQRRRGGSarGVQGLVHISEI-NH_2_ (R23EI) (Figure 3).

The N-terminus of synthesized hybrid peptides contains a cell-penetrating peptide (CPP) fragment RKKRRQRRR connected to the amyloidogenic region using a linker of three glycine and sarcosine residues (GG-Sar-G) [29]. In addition, the R23F peptide was synthesized with an amino acid modification (Asi19 instead of Asn19). Amino acid residues Sar and Asi were added to the structure of peptides, as they allow obtaining a more stable conformation of the final product of peptide synthesis. Unfortunately, AlphaFold 2 does not work with Sar and Asi. In this regard, we used analogs for calculations: proline (P) and alanine (A) instead of Sar for all peptides, and Asn instead of Asi for only R23F. AlphaFold 2 allows you to model the structure of the peptide as a whole and show individual elements of an ordered structure, for example, α-helices or β-strands, which are indicated in Figure 3 by helices and arrows, respectively. The absence of such ordered elements of the secondary structure indicates, accordingly, the disordered structure of the peptide. As can be seen from Figure 3, the R23F, R23DI, and R23EI peptides are predicted to be predominantly disordered. At the same time, the AlphaFold 2 program predicts short β-strands and α-helix at the C-terminus of the R23F and R23DI peptides, respectively. The common structural property of all models of the R23F, R23DI, R23EI peptides linked to the N-terminal sequence is the low tendency for oligomerization of the RKKRRQRRRGGSarG region. At the same time, the formation of compact ordered regions is characteristic of the C-terminal amyloidogenic sequence peptides. This duality in the structure of peptides may contribute to their antimicrobial properties (Appendix A). Thus, Appendix A shows that the RKKRRQRRRGGSarG sequence is not predicted to be amyloidogenic. At the same time, based on the data in Appendix A, it can be concluded that the RKKRRQRRRGGSarG sequence makes a significant contribution to the positive charge of hybrid peptides R23F, R23DI, R23EI as well as to the results of predicting these peptides as AMPs. Based on the models predicted by AlphaFold 2, it can be assumed that the probability of oligomer formation for R23F and R23EI is higher than for R23DI. Oligomers for R23F and R23EI are more likely than for R23DI since they retain predicted ordered structures.

### 2.3. Determination of the Antibacterial Activity of the R23F, R23DI, R23EI Peptides by Agar Diffusion Assay

At the next stage, the predicted antimicrobial properties of the R23F, R23DI, and R23EI peptides were tested using experiments on agar with cells of MRSA (ATCC 43300 strain), *S. aureus* (209P strain), *P. aeruginosa* (ATCC 28753 strain), *E. coli* (K12 strain), and *B. cereus* (IP-5812 strain) (Figure 4). The antibacterial action of gentamicin sulfate on the cell culture growth served as a positive control.

*B. cereus* cells were found to be the most sensitive to peptides. Thus, for the R23F, R23DI, and R23EI peptides, the value of the minimum inhibitory concentration (MIC) was 75 µM, 150 µM, and 300 µM, respectively. Among the individual peptides, the highest antibacterial activity was found for the R23F peptide (MIC was 300 µM against *S. aureus* and *E. coli*) and the lowest for R23EI (MIC ≥ 300 µM against all using microorganisms). For the R23DI and R23EI peptides used against *P. aeruginosa*, more growth of strain cells was observed at the areas of peptide preparations. Perhaps, instead of suppressing the growth of *P. aeruginosa* bacterial cells, the opposite effect was observed. The least antibacterial effect was in tests against MRSA (MIC ≥ 300 µM for all testing peptides).

### 2.4. Antibacterial Activity of Peptides against MRSA, S. aureus, and P. aeruginosa in Liquid Medium

We tested peptides R23F, R23DI, and R23EI in a liquid medium with MRSA cells (ATCC 43300 strain) (Figure 5). However, in this experiment, in contrast to those presented above, the antimicrobial effects of R23F, R23DI, and R23EI were detected during 16 h of co-incubation of the peptides with MRSA cells.

**Figure 4 ijms-23-00524-f004:**
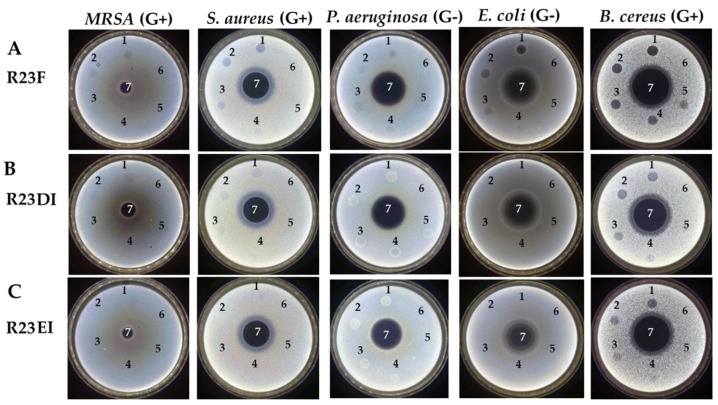
Results of testing the antimicrobial effects of peptides R23F, R23DI, R23EI against MRSA (Gram-positive, G+), *S. aureus* (Gram-positive, G+), *P. aeruginosa* (Gram-negative, G−), *E. coli* (Gram-negative, G−), and *B. cereus* (Gram-positive, G+). Bacteria were grown on solid agar medium LB. Application scheme: 1—peptide 300 μM; 2—peptide 150 μM; 3—peptide 75 μM; 4—peptide 37.5 μM; 5—peptide 18.75 μM; 6—LB with DMSO 2% (*v*/*v*); 7—gentamicin sulfate 1700 μM.

**Figure 5 ijms-23-00524-f005:**
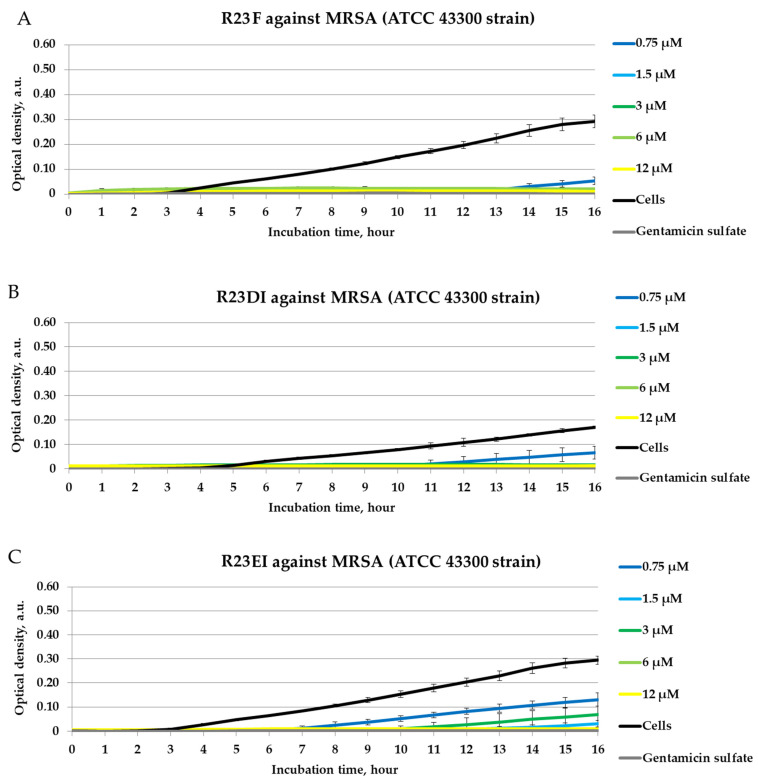
Inhibitory activity of peptides R23F (**A**), R23DI (**B**), R23EI (**C**) during 16 h of incubation with MRSA strain ATCC 43300 cells. A gentamicin sulfate concentration of 1700 μM was used as a positive control of the antimicrobial effect. ATCC 43300 cells without peptide were used as negative controls. The experiments were performed in three replicates. The results were expressed as the mean ± standard deviation (M ± SD).

Inhibition of MRSA growth for 24 h co-incubation with peptides R23F (1.5–12 μM, MIC was 6 μM), R23DI (1.5–12 μM, MIC was 6 μM), and R23EI (6–12 μM, MIC was 12 μM) was found. It should be noted that the peptides, by inhibiting the growth of MRSA cells, may have a temporary antibacterial effect, which tends to decrease over time.

The antimicrobial action of the peptides was also assessed against cells of the *S. aureus* 209P strain and *P. aeruginosa* ATCC 28753 strain (Figure 6 and Figure 7).

The effects of inhibition of *S. aureus* (strain 209P) and *P. aeruginosa* (strain ATCC 28753) cell growth were revealed for R23F (MIC was 12 μM) and R23DI (MIC was ≥12 μM). The lowest antibacterial activity was found for the R23EI peptide (MIC was >12 μM).

Similar experiments on testing peptides R23F, R23DI, and R23EI have been performed against other strains MRSA, *S. aureus*, and *P. aeruginosa* cells. Figure 8 shows the results of the peptide test against MRSA SA 180-F strain, *S. aureus* 129B strain, and *P. aeruginosa* PA103 strain cells in liquid LB medium after 24 h co-incubation.

MRSA strain ATCC 43300 (Figure 5) and strain SA 180-F (Figure 8A) cells differed in their growth rate, as well as *S. aureus* strain 209P (Figure 6) and strain 129B (Figure 8B) cells. MRSA strain ATCC 43300 and *S. aureus* strain 209P grow more slowly than MRSA strain SA 180-F and *S. aureus* strain 129B respectively, which is reflected in the lower measured optical density of the samples. As shown in Figure 8, after 24-h co-incubation of cells with peptides, as a rule, R23F, R23DI, and R23EI do not inhibit the growth of MRSA (SA 180-F strain), *S. aureus* (129B strain), and *P. aeruginosa* (PA103 strain) in the range of tested concentrations from 0.75 μM to 375 μM (MIC was for all testing peptides > 375 μM). Optical density decreases significantly only during incubation of 375 μM R23F with MRSA (SA 180-F strain) and *S. aureus* (129B strain) (Figure 7A,B).

### 2.5. Viability of MRSA and S. aureus Cells after Peptide Treatment

To clarify the bactericidal or bacteriostatic effect of peptides R23F, R23DI, and R23EI, we tested the survival of MRSA using SYTO 9 (emissions from 505–515 nm) and propidium iodide (PI) (emissions from 600–610 nm) dyes. The data were visualized using two-dimensional fluorescence microscopy (Figure 9 and Figure 10).

For the R23F peptide, as for the R23DI peptide, when a concentration of 0.75 μM is used, the growth of bacterial cells is inhibited as shown in Figure 9. The cells of the MRSA (ATCC 43300 strain) change shape at a concentration of 3 μM R23DI peptide, which indicates a violation of the cell wall. The same experiments were performed on *S. aureus* cells treated with the same peptides (Figure 10).

*S. aureus* cells were destroyed after co-incubation with peptide R23F or R23DI at a concentration of 3 μM, but the shape of the cells does not change. The MRSA ATCC 43300 strain is more sensitive to the R23DI peptide compared to the *S. aureus* cells of the 209P strain. The R23EI peptide shows lower antimicrobial activity, from the graphs and the live or dead cell staining test, it can be seen that higher inhibitory concentrations are required, but this peptide is more effective against the MRSA strain ATCC 43300. When cells are tested with the R23EI peptide, the cell shape remains unchanged, which indicates an antimicrobial effect without compromising the integrity of the membrane.

**Figure 9 ijms-23-00524-f009:**
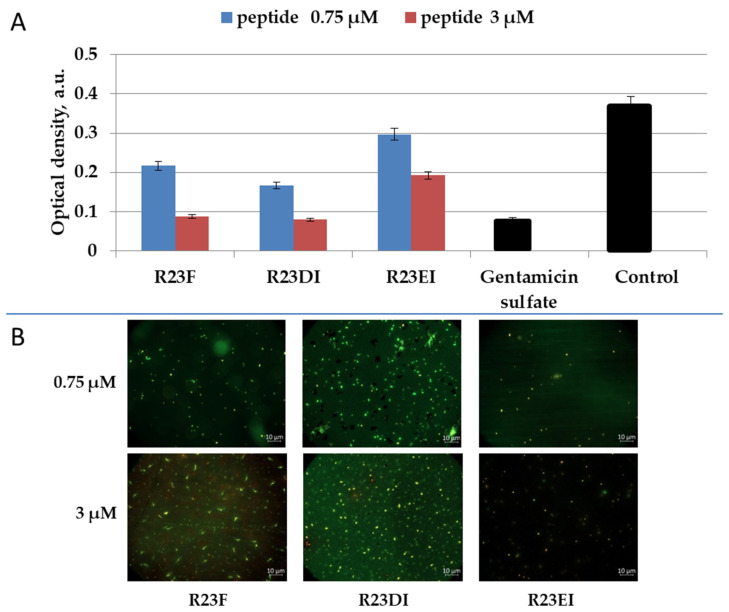
Antimicrobial effect of peptides on MRSA cells (ATCC 43300 strain) after one day of co-incubation (**A**). Fluorescence microscopic images of MRSA cells (ATCC 43300 strain) treated with peptides R23F, R23DI, and R23EI (**B**). A gentamicin sulfate concentration of 1700 μM was used as a positive control for the antimicrobial effect. Bacterial cells stained with SYTO 9 and propidium iodide, but fluorescence signal of SYTO 9 is visible only since the cells are not dead, but growth of bacterial cells is inhibited. The experiments were performed in two replicates. The results were expressed as the mean ± standard deviation (M ± SD).

**Figure 10 ijms-23-00524-f010:**
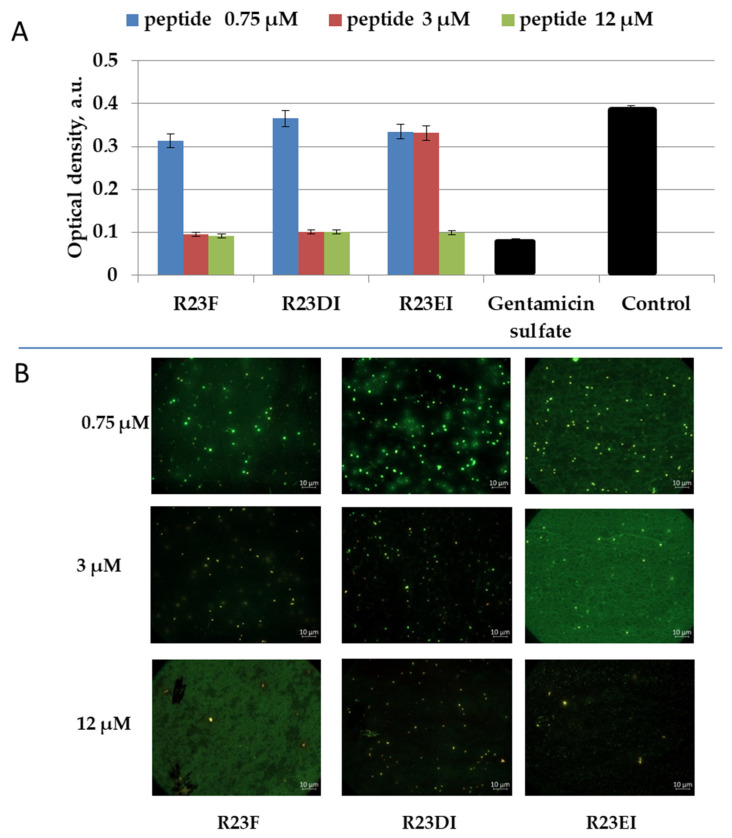
Antimicrobial action of peptides against *S. aureus* cells (209P strain) (**A**). Fluorescence microscopic images of *S. aureus* cells (209P strain) treated with peptides R23F, R23DI, and R23EI (**B**). A gentamicin sulfate concentration of 1700 μM was used as a positive control for the antimicrobial effect. Bacterial cells stained with SYTO 9 and propidium iodide, but fluorescence signal of SYTO 9 is visible only since the cells are not dead, but growth of bacterial cells is inhibited. The experiments were performed in two replicates. The results were expressed as the mean ± standard deviation (M ± SD).

Other strains MRSA and *S. aureus* were also investigated using live/dead staining. Representative fluorescence images of cells after 24 h incubation in the presence and absence of peptides are shown in Figure 11.

The shape of the cells is not changed after co-incubation with peptide R23F, however, the amount of cells in the field of view in the sample with a concentration of 375 μM is lower than in the sample with a concentration of 37.5 μM in both strains. When cells were tested with R23DI and R23EI, no differences in the number and shape of cells were detected during microscopy of samples with peptide concentrations of 37.5 μM and 375 μM. In general, the results of the bacterial viability test for MRSA (SA 180-F strain) and *S. aureus* (129B strain) correspond to the data on optical densities of the samples (Figure 7A,B) and confirm that the strains low susceptibility to the peptides.

### 2.6. Toxicity of R23F, R23DI, and R23EI against Eukaryotic Cells

Since peptides R23F, R23DI, and R23EI can inhibit growth and cause the death of bacterial cells, it was necessary to check their cytotoxic effect on eukaryotic cells. To solve this problem, we tested hybrid peptides against cell lines of human fibroblasts and human breast tumor cells (Figure 12).

It was shown that the peptides with the increase in concentration, both with and without preliminary incubation (18 h, 37 °C, in DMEM with 10% FBS), could suppress the viability of fibroblasts and human breast tumor cells (BT-474). The analysis of the cytostatic action was carried out based on the distribution of cells by the phases of the cell cycle and mitotic activity (Table 1).

Incubation of BT-474 cells with peptides at a concentration of 20 μM did not change the cell proliferative activity. In addition, peptides at the same concentration did not modify the migration activity of fibroblast and BT-474 cells (data not shown).

## 3. Discussions

In our study, we found that peptides synthesized based on the ribosomal S1 protein sequence from *S. aureus* have significant antimicrobial activity against *Staphylococcus aureus*, MRSA, *Bacillus cereus*, *Escherichia coli*, and *Pseudomonas aeruginosa*, and their antimicrobial effect is manifested at the level of the antibiotic gentamicin sulfate. The antimicrobial effect of the peptides was higher against *S. aureus*, MRSA, and *P. aeruginosa* cultured in liquid medium compared to the effect against pathogens grown on agar. Unexpectedly, hybrid peptides showed a significant effect against *B. cereus* but not against *S. aureus* and MRSA cells grown on agar (Figure 4). At the same time, for all peptides, MIC was 12 μM for testing under liquid conditions against MRSA (strain ATCC 43300) and *S. aureus* (strain 209P) (Figure 5 and Figure 6).

The high amyloidogenicity of the S1 protein from *S. aureus* was largely due to the high amyloidogenicity of the fourth domain, which was predicted by all four programs: FoldAmyloid [14], AGGRESCAN [15], Pasta 2.0 [16], and Waltz [17] (Figure 2). Although domain 3 (D3) is predicted to be the most amyloidogenic for 1453 S1 proteins and with the greatest identity, it should be noted that domain 4 (D4) is more amyloidogenic than D3 for *S. aureus* ribosomal S1 protein. Based on the S1 proteins from two model organisms (*T. thermophilus*, *E. coli*) and two pathogenic organisms (*P. aeruginosa*, *S. aureus*), we synthesized at least one amyloidogenic peptide from the third domain [27,42,43]. In this regard, the propensity of some proteins and peptides to form oligomers without fibril formation, as expected, can be used to predict their amyloidogenicity, as was suggested [22]. Previously, it was shown that the formation of oligomers is a key event that determines the pathway of fibrillation [23,44,45], as well as a polymorphism of fibrils [46,47,48].

A preliminary analysis of the antimicrobial effects of short peptides demonstrated that only two peptides, from the first and the fourth domain, have a slight antimicrobial effect (see Appendix A). Moreover, since the third domain is the most conserved, a hybrid peptide (R23DI) was also synthesized based on the LTQFGAFIDI sequence from D3. It should be noted that the LTQFGAFIDI peptide is similar to the peptide with amyloidogenic and antimicrobial properties synthesized based on the amyloidogenic sequence of ribosomal S1 protein from *Thermus thermophilus* (V10I) [27].

Previously, the free-energy landscape was used to model the pathways of protein folding and peptides, and the most common descriptor was population density in some chosen type of projection of the configuration space [49,50]. To evaluate how the overall peptide structure related to their ability to form oligomers, we built three-dimensional models of the peptides. Using the recently published program AlphaFold 2 [41], we predicted 3D models for monomers, trimers, and tetramers of the R23F, R23DI, and R23EI peptides (Figure 3). Despite the significant success of AlphaFold 2 in predicting the tertiary structure of proteins, its use for modeling the structure of multi-domain proteins (such as ribosomal S1 protein from *S. aureus*), intrinsically disordered regions, and amyloidogenic peptides has restrictions [51,52,53].

Testing of the antimicrobial peptides R23F, R23DI, and R23EI on agar against Gram-positive (MRSA, *S. aureus*, and *B. cereus*) and Gram-negative (*E. coli* and *P. aeruginosa*) bacteria revealed multiple effects of the peptides (Figure 4). On the one hand, peptides R23F, R23DI, and R23EI suppressed the growth of *B. cereus.* On the other hand, peptides R23DI and R23EI did not have a significant antimicrobial effect against MRSA, *S. aureus*. Moreover, for R23DI and R23EI was observed the opposite effect in relation to *P. aeruginosa* cells grown on agar. The antibacterial effects R23F, R23DI, and R23EI against the strains ATCC 43300 MRSA, 209P *S. aureus,* and ATCC 28753 *P. aeruginosa* (Figure 5, Figure 6 and Figure 7) are more pronounced compared to the test results against strains SA 180-F MRSA, 129B *S. aureus*, and PA103 *P. aeruginosa* (Figure 8)*,* respectively. Such strain differences in sensitivity to the action of antimicrobial peptides R23F, R23DI, and R23EI corresponded to the differences of these strains to the action of traditional antibiotics (see Section 5.3 “Microorganisms”). These results may be related to the strain-specific resistance of bacteria to the action of antimicrobial agents [54,55]. The peptides R23F, R23DI, and R23EI demonstrated MIC around 12 μM against individual strains of MRSA and *S. aureus* indicating their antimicrobial activity at the level of peptide antibiotic polymyxin B [56]. The antimicrobial effect of the peptides against other strains and pathogenic microorganisms can be increased due to amino acid modifications of the sequence of antimicrobial peptides, for example as was done for jelleine-1 [57]. Based on the experiments carried out on the effect of the R23DI peptide on *S. aureus* and MRSA, it can be concluded that the R23DI peptide disrupts the cell wall of MRSA strain ATCC 43300 cells (Figure 9) and does not destroy the cell wall when exposed to the *S. aureus* strain 209P cells (Figure 10). The results of the bacterial viability test for MRSA strain SA 180-F cells and strain 129B *S. aureus* cells (Figure 11) obtained with the help of two-dimensional fluorescence microscopy correspond to the data of previous experiments that these strains are less sensitive to the action of the peptides. The action of hybrid antimicrobial peptides R23F, R23DI, R23EI is not limited to only one mechanism, but may be cell wall permeabilization or cytoplasmic membrane depolarization. In general, the combination of amino acid regions with different properties determines a synergistic effect or several mechanisms of action of hybrid peptides [58], which ultimately manifests itself in a wide spectrum of their action against various types of bacteria. Antimicrobial peptides, in addition to antibacterial activity, can exhibit various biological effects, for example, antidiabetic action [59], anti-cancer effects of some AMPs have been shown [60,61]. It is known that minor changes in the primary structure of AMP can significantly affect not only the antimicrobial efficiency of peptides but also their mechanism of action [29,62].

## 4. Conclusions

In this study, we have developed and synthesized antimicrobial peptides R23F, R23DI, R23EI. We have demonstrated that hybrid peptides R23F, R23DI, and R23EI based on the ribosomal S1 protein sequence from *S. aureus* can be used as antimicrobial peptides against Gram-positive and Gram-negative bacteria. The fusion of amyloidogenic regions of the protein with part of the cell-penetrating peptide (e.g., TAT peptide [63]) opens up new possibilities for the manifestation of the antimicrobial effects of hybrid peptides through several mechanisms. In this regard, such hybrid peptides may be more effective in the long term against a wide range of bacteria, preventing them from developing resistance to treatment.

## 5. Materials and Methods

### 5.1. Bioinformatics Analysis of Peptides

Amyloidogenic regions of protein S1 from *S. aureus* with a length of at least five amino acid residues were determined using default software setting FoldAmyloid [14], AGGRESCAN [15], Pasta 2.0 [16], and Waltz [17]. FoldAmyloid available online: http://bioinfo.protres.ru/fold-amyloid/ (accessed on 3 August 2021). AGGRESCAN is available online: http://bioinf.uab.es/aggrescan/ (accessed on 12 August 2021). Pasta 2.0 is available online: http://old.protein.bio.unipd.it/pasta2/ (accessed on 5 August 2021). Waltz is available online: http://waltz.switchlab.org/index.cgi (accessed on 5 August 2021).

### 5.2. Synthesis and Characterization of Peptides

#### 5.2.1. Peptide Synthesis

Peptides VVVHINGGKF (V10F), GVVVRLANFG (G10G), VQGLVHISEI (V10I), QQVNVKILGI (Q10I) were commercial products (IQ Chemical LLC, S. Petersburg, Russia). Solid-phase synthesis of peptides RKKRRQRRRGGSarGVVVHI-Asi-GGKF-NH_2_ (R23F), RKKRRQRRRGGSarGLTQFGAFIDI-NH_2_ (R23DI), and RKKRRQRRRGGSarGVQGLVHISEI-NH_2_ (R23EI) was performed by Boc/Bzl methodology [64] using a standard set of protected amino acid derivatives and TBTU as a coupling reagent [65,66] starting with MBHA resin. The completeness of the acylation reactions was monitored using Kaiser’s ninhydrin test [67]. Aspartic acid residue side chain in peptide R23DI was protected as cyclohexyl ester, while in peptide R23F OBzl protecting group was used in order to facilitate cyclization of aspartic acid residue into aminosuccinimide. Peptide deprotection, cleavage from the solid support as well as aminosuccinimide formation [68] in R23F were performed using 1M TFMSA/thioanisole in TFA at 25 °C for 2 h [69,70]. Crude peptides were precipitated with anhydrous ether, dried in vacuo over KOH pellets, and purified by gel filtration (Sephadex G-10) followed by semi-preparative HPLC in isocratic mode (mobile phase “A” 0.1% TFA in water, mobile phase “B” acetonitrile (no additives)) on Luna C18 250 × 21.5 mm (10 μm) column (Phenomenex, Torrance, CA, USA) at flow rate 10 mL/min. The collected fractions were analyzed using RP-HPLC on Luna 5 µm C18 (2) 100 Å 250 × 4.6 column (Phenomenex, Torrance, CA, USA). The HPLC profiles of the synthetic peptides are presented in Appendix A. The conformity of synthesized peptide sequences was determined by mass spectrometric analysis using an Orbitrap Elite mass spectrometer (Thermo Scientific, Dreieich, Germany).

#### 5.2.2. Prediction of Secondary Structures of Peptides by AlphaFold 2

We performed modeling of folding patterns for hybrid peptides R23F, R23DI, and R23EI, using AlphaFold 2 [41]. The source code of AlphaFold 2 was retrieved from the GitHub repository (https://github.com/deepmind/alphafold) on 8 September 2021. For all peptides, three models were generated.

### 5.3. Microorganisms

Two MRSA strains were used: a sensitive strain ATCC 43300 (resistant only to ampicillin) and clinical isolate strain SA 180-F (resistant to benzylpenicillin, oxacillin, erythromycin, clindamycin, ciprofloxacin, vancomycin, sulfamethoxazole, and levomycetin). Two *S. aureus* strains were used: a sensitive strain 209P and clinical isolate strain 129B (resistant to benzylpenicillin, oxacillin, erythromycin, clindamycin, vancomycin, and sulfamethoxazole) Two P. aeruginosa strains were used: a sensitive strain ATCC 28753 and resistant strain (resistant to sulfamethoxazole).

### 5.4. Determination of the Antibacterial Activity of Peptides by Agar Diffusion Assay

The preparation and co-incubation of peptides with cells MRSA (ATCC 43300 strain), *S. aureus* (209P strain), *P. aeruginosa* (ATCC 28753 strain), *E. coli* (K12 strain), and *B. cereus* (IP-5812 strain) were carried out as described previously [29]. The strains were inoculated with a wire loop in 5 mL of Luria-Bertani (LB) medium and were incubated at 37 °C for 12 h. After 12 h, 50 μL of an overnight culture of the test bacteria was added to each dish, plastic dishes were poured with LB medium (5 mL) with 0.75% agar. The R23F, R23DI, and R23EI peptides were finally dissolved in 2% (*v/v*) DMSO and were tested with different concentrations. The plates with the medium were dried and 10 μL of the prepared peptide was applied on top of solid agar LB medium. Then, the dishes were placed in a thermostat and incubated for 12 h at a temperature of 37 °C. Antimicrobial activity was recorded by the presence of transparent zones of no growth around drops with peptides. 1700 μM gentamicin sulfate was used as control of the antimicrobial effect.

### 5.5. Measurement of the Antibacterial Activity of Peptides against MRSA, S. aureus, and P. aeruginosa in Liquid Medium

The R23F, R23DI, and R23EI peptides and gentamicin sulfate were dissolved in 100% DMSO and solutions were used with a final DMSO concentration of 2% (*v/v*) to measurement their activity. Determination of the growth for cells MRSA strain ATCC 43300, *S. aureus* strain 209P, *P. aeruginosa* strain ATCC 28753 was performed using Mueller-Hinton Broth (MHB) (Sigma-Aldrich, St. Louis, MO, USA). Cells of MRSA (strain SA 180-F) and *S. aureus* (strain 129B) were grown on the brain-heart infusion (BHI) medium, cells of *P. aeruginosa* (strain PA103) were grown on the LB medium. The preparation and co-incubation of peptides with cells were carried out as described previously [29].

### 5.6. Assessment Bacterial Viability

The Thermo Scientific kit (№ L7012, 1871164, and 1939603) was used for the cell test LIVE/DEAD. The LIVE/DEAD differentiates live and dead cells using membrane integrity as a proxy for cell viability and is based on a dual staining procedure using SYTO 9 (emissions from 505–515 nm) and propidium iodide (PI) (emissions from 600–610 nm). When using the test, cells are stained in the appropriate colors, namely, green indicates the living cells, yellow color, turning into red, indicates a violation of the vital functions of cells. Working solutions of SYTO 9 and PI were established as described previously in the article [71]. Fluorescence measures are typically taken using microscopes Zeiss Axio Scope. A1 and Zeiss Axio Imager M2 (Carl Zeiss Microscopy GmbH, Hamburg, Germany).

### 5.7. Determination of Toxicity of Peptides against Eukaryotic Cells

Cell viability, distribution of cells by the phases of the cell cycle, and mitotic activity were estimated as described previously [27]. Each assay was performed in triplicate. All measurements were carried out on the control samples that were not treated with R23F, R23DI, and R23EI peptides.

### 5.8. Statistical Analysis

A SigmaPlot 14.0 software package (SPSS 14.0, SPSS Inc., Chicago, IL, USA) was used for the statistical analysis. The results were expressed as the mean ± standard deviation (M ± SD). The experiments were performed in at least two replicates (*n* ≥ 3). The statistical significance of the difference was determined using analysis of variance (ANOVA) and the Student’s *t*-test.

## Figures and Tables

**Figure 1 ijms-23-00524-f001:**
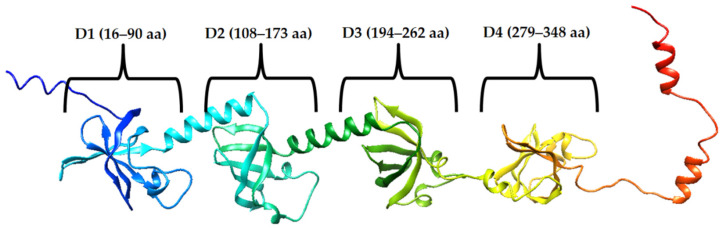
Schematic representation of the domain organization (D1, D2, D3, and D4) of the ribosomal S1 protein from *S. aureus*. The S1 sequence is taken from the UniProt database (UniProt. Available online: https://www.uniprot.org/uniprot/Q6G987 (accessed on 19 August 2021)). The tertiary structure of S1 was predicted by AlphaFold 2 [41] (AlphaFold 2. Available online: https://github.com/deepmind/alphafold (accessed on 8 September 2021).

**Figure 3 ijms-23-00524-f003:**
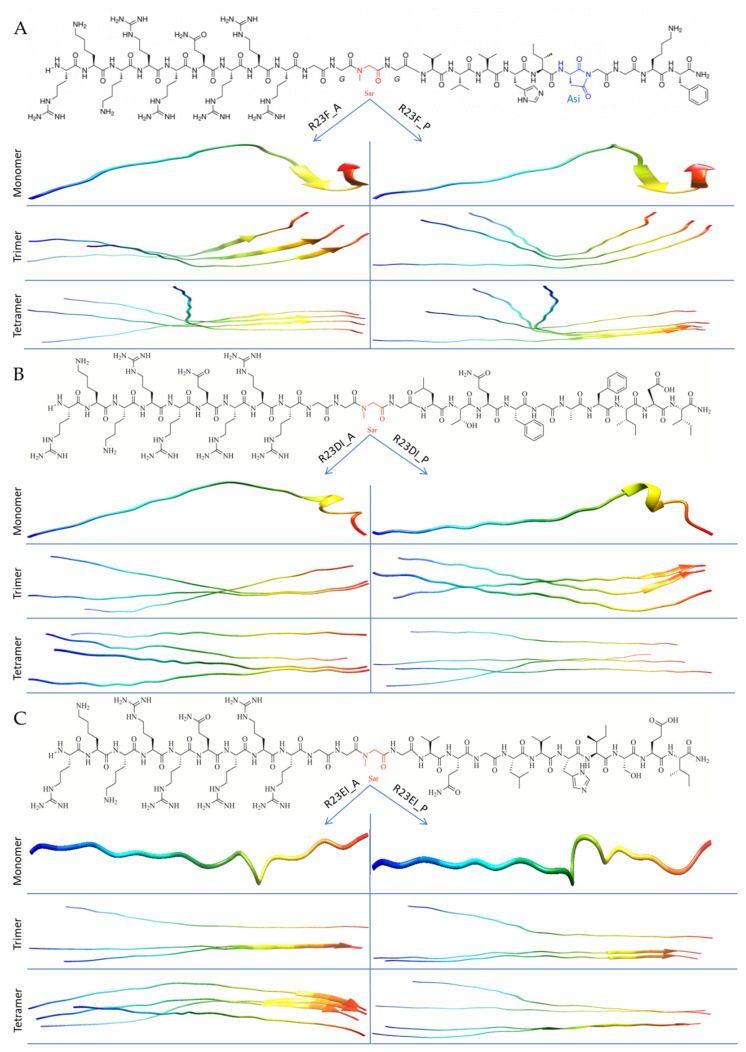
Primary structure and folding patterns of hybrid peptides R23F (**A**), R23DI (**B**), and R23EI (**C**) were predicted by AlphaFold 2. In the primary structure of the R23F peptide, a modified asparagine residue (aminosuccinimide, Asi) is shown in blue, and sarcosine residues (Sar) are shown in red for the three peptides. To model the structure of peptides using AlphaFold 2, sarcosine residues were replaced with alanine (A) or proline (P) residues in the left and right portions of the figure correspondently [41].

**Figure 6 ijms-23-00524-f006:**
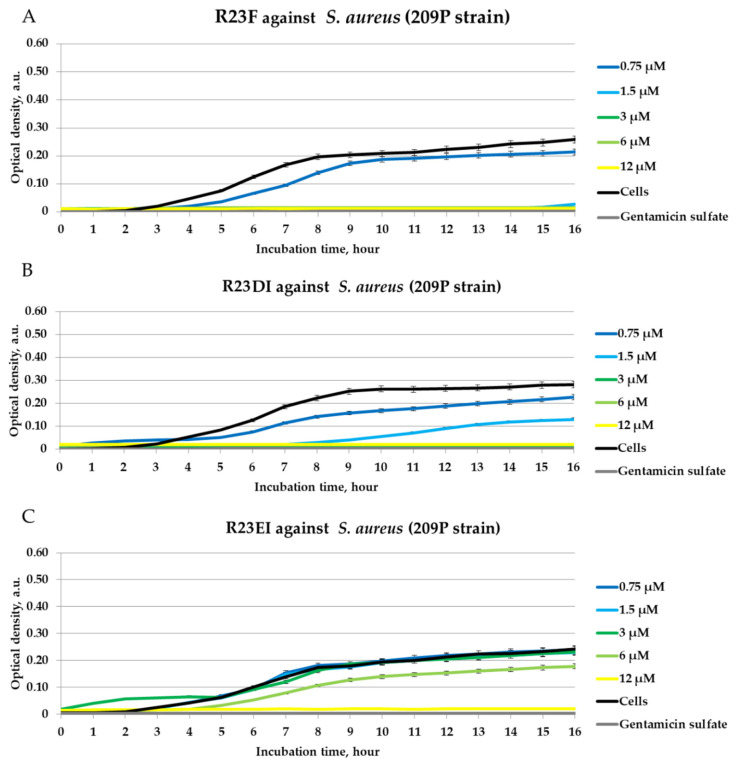
Inhibitory activity of peptides R23F (**A**), R23DI (**B**), R23EI (**C**) during 16 h of incubation with *S. aureus* strain 209P cells. A gentamicin sulfate concentration of 1700 μM was used as a positive control of the antimicrobial effect. 209P cells without peptide were used as negative controls. The experiments were performed in three replicates. The results were expressed as the mean ± standard deviation (M ± SD).

**Figure 7 ijms-23-00524-f007:**
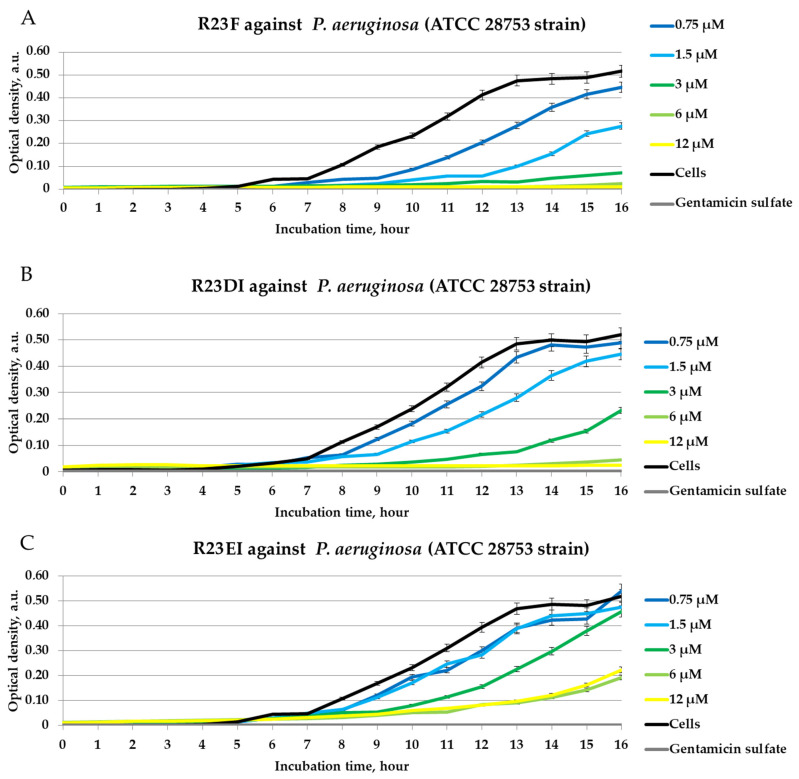
Inhibitory activity of peptides R23F (**A**), R23DI (**B**), R23EI (**C**) during 16 h of incubation with *P. aeruginosa* strain ATCC 28753 cells. A gentamicin sulfate concentration of 1700 μM was used as a positive control of the antimicrobial effect. ATCC 28753 cells without peptide were used as negative controls. The experiments were performed in three replicates. The results were expressed as the mean ± standard deviation (M ± SD).

**Figure 8 ijms-23-00524-f008:**
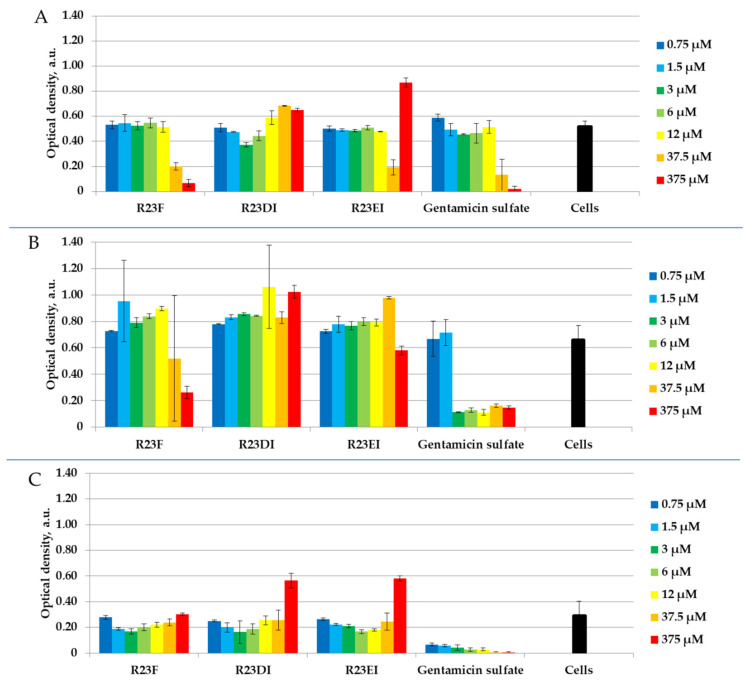
Antibacterial activity of peptides R23F, R23DI, and R23EI after 24 h co-incubation with MRSA (SA 180-F strain) (**A**), *S. aureus* (129B strain) (**B**)*,* and cells the *P. aeruginosa* (PA103 strain) (**C**). For comparison with other strains see previous Figure 5 (MRSA strain ATCC 43300), Figure 6 (*S. aureus* strain 209P), and Figure 7 (*P. aeruginosa* strain ATCC 28753). The experiments were performed in two replicates. The results were expressed as the mean ± standard deviation (M ± SD).

**Figure 11 ijms-23-00524-f011:**
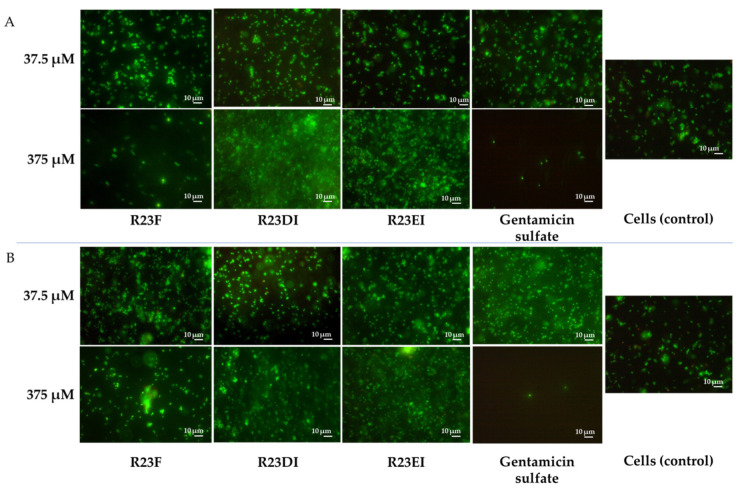
Fluorescence microscopic images of (**A**) MRSA (SA 180-F strain) and (**B**) *S. aureus* (129B strain) cells treated with peptides R23F, R23DI, and R23EI.

**Figure 12 ijms-23-00524-f012:**
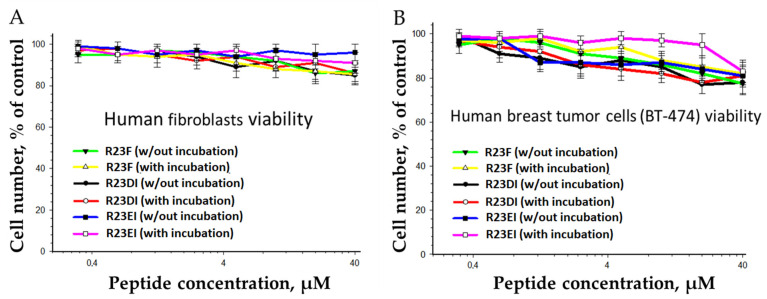
Effect of peptide treatment on the survival of human fibroblasts (**A**) and the breast tumor cell line BT-474 (**B**) without preliminary incubation (18 h, 37 °C, in DMEM with 10% FBS) and after 72 h of co-incubation with peptide. Each of the experiments was carried out at least three times (*n* ≥ 3). Error bars show standard deviation (SD).

**Table 1 ijms-23-00524-t001:** Results of cell distribution by phases of the cell cycle after peptide treatment.

	G1 Phase	S Phase	G2 Phase	M Phase
Peptides + BT-474	73 ± 5%	16 ± 3%	7 ± 2%	4 ± 1%
Control (BT-474)	72 ± 5%	20 ± 3%	5 ± 1%	3 ± 1%
Peptides + fibroblasts	72 ± 4%	17 ± 3%	8 ± 2%	3 ± 1%
Control (fibroblasts)	78 ± 5%	13 ± 2%	7 ± 2%	2 ± 1%

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
