# Peer review of "Multiple Antimicrobial Effects of Hybrid Peptides Synthesized Based on the Sequence of Ribosomal S1 Protein from Staphylococcus aureus"

_ijms, 2022, doi:10.3390/ijms23010524_

Round 1

Reviewer 1 Report

The authors aim to help solve the growing problem of the waning effectiveness of traditional antibiotics by developing new anitimicrobial peptides with amyloidogenicity. Their approach is to create hybrid peptides with fragments of cell-penetrating peptides and protein regions prone to the formation of ordered aggregates. The study design includes testing against a variety of microorganisms that differ in important ways to show how universal their approach can be, as well as investigating potential toxic effects on eukaryotic cells. I was excited to read this paper to see how this field is progressing, but I found that there are several major flaws in this study.

Major comments:

  • Why was the ribosomal S1 protein of S. aureus specifically chosen for your designs, rather than the same (or a different) protein from a different organism? The rationale for the specific organism is missing, given you compared 25 different phyla.

  • Do you trust any of the bioinformatics predictions of amyloidogenicity more than the others, before or after this study?

  • It can be interpreted, but it would be helpful if you explained in the figure legend how your color coding scheme of Figure 2B relates to Figure 2A, as well as why some of these regions are in bold.

  • Please explain exactly what “preliminary analysis” of antimicrobial action you performed to down-select your peptides, referred to on page 4 line 141. Could be shown in supplemental.

  • Before you added the peptide from D3 to the study in line 145, I had written the question: “Despite the predictions of the various programs, why didn’t you include a peptide from the D3 region if that region has “the highest identity and amyloidogenicity”. The peptide that was found by both FoldAmyloid and AGGRESCAN might be interesting.” I would suggest including a box around that 5th peptide in Figure 2 to simplify your results section, even if including it was an afterthought. Did you run the same preliminary analysis of antimicrobial action on L10I, or just synthesize a hybrid peptide with it?

  • On line 156, please explain why you used the modified asparagine residue “Asi” (aminosuccinimide) instead of asparagine (Asn, or N), and define it in the main text. It’s not sufficient to only define it in Figure 3.

  • Line157: Figure 2 does not show the structure of the peptides, and you say it shows they are disordered. Please correct this statement and be clear on how exactly you are comparing the peptides and coming to the conclusions you make about their structure. Are you referring to Figure 3? Please do not assume your readers are familiar with AlphaFold 2. What does the color from N to C terminus indicate, if anything? What is the difference between the data on the left and right sides of the figure (peptide_A vs peptide_P)? Due to the arrows coming from the chemical structure, it seems as if you are indicating that the left vs right sides of the figure are the structures of the left and right portions of the peptide with respect to the Sar residue, but you do not clarify in the legend or text. Furthermore, the structures look too similar on the left and right sides of this figure to be accurate if these two halves of the peptides were analyzed separately here. I believe that the left and right sides are showing the same full peptide structure with different settings based on your statements and conclusions about their structure, but please explain this figure and your conclusions more sufficiently in the text. What exactly about the models shows that the probability of oligomer formation is higher for R23F and R23EI than for R23DI?

  • Line 164: Please provide further explanation of what is shown in the supplementary tables to support your claims.

  • Line 183: What do you mean by “unusual growth zones”, can you describe this more clearly? Was there MORE growth in those areas, or unusual looking cells?

  • Why did you test the antimicrobial peptides at such different concentrations in solid (18.75 to 300 uM in Figure 5) vs liquid media (0.75 to 12 uM in Figure 6)? Typically, its more common to reduce the concentration of antibiotic when culturing on agar, so is there a reason this is different with the peptides? Was there some trial and error involved in determining these concentrations? If you are following prior protocols giving precedent, please cite them. Did you test more concentrations than you show here, or did you know that you did not need to go as high in concentration to see an effect from literature or previous studies? This is especially of interest because you test up to 375 um in liquid culture in Figure 7, where it seemed necessary.

  • 2% v/v DMSO was present in the tests on agar due to solubilization of the peptides. How much DMSO was present in the liquid tests, and was it controlled for in some way across the samples?

  • Line 207: Was 16 hours the optimal timing for the peptides? Because of the increase in growth starting at 16 hours, it would be nice to see a growth curve for P. aeruginosa similar to the one for MRSA in Figure 5. It sounds like you did collect that data.

  • Figures 6 and 7 would be easier to compare if you kept the coloring consistent for the concentrations tested across both.

  • You should explain what is different in the experiments performed in Figure 5 vs Figure 7A for MRSA in the text to make it easier to follow. You note in the figure legends that the strain in Figure 5 was ATCC 43300 and in Figure 7A was SA 180-F. Is the SA 180-F strain known to be more resistant to antibiotics, faster growing, or more aggressive in any way to explain this discrepancy in effectiveness of the peptides? Why did you choose these strains (are there others you excluded)? Beyond just listing the names, pointing out more explicitly that a different strain is tested in Figure 7A vs Figure 5, without leaving the reader to interpret, would be helpful.

  • Figures 8 and 9: It’s not clear if the stained cells in B are from the exact same cultures shown in A, please clarify. It could make a difference in terms of the total number of cells seen in the image field. Was there an effort to normalize by cell number in B, or did you just put a set volume on a slide? Why are there less cells in the field of view for R23EI when the optical density compared to the other peptides was higher in Figures 8A and 9A?

  • Figures 8 and 9: Images of the gentamycin control would be good to show here for comparison, like you do in Figure 10. You should run these tests with both SYTO-9 and propidium iodide so we can see cells that are present but dead. You mention using PI in your methods, but not here. If you included PI in the study but all cells survived and appear green, please state so. In that case, it would be helpful to see the red field alone (even if supplemental, you have a lot of figures). Please mention in general that you stained with SYTO-9 (and PI?) in the figure legends, and what magnification was used. Thank you for the scale bars.

  • Microscope image quality in general is inconsistent. There are circular edges. Is this an artifact of a microscope setting (since it’s in the same location in each image)? Why does the background look like a general green films is present in some of these (such as Figure 9B, 3 um R23EI).

  • Line 232: I can’t distinguish a change in shape at this magnification, the cells are too small. Can you show what you are describing at a higher magnification? You also talk about cell shape in lines 240, 245, and 253: same issue. Changes in shape are not convincing at this magnification. Staining with propidium iodide would also be useful here for any claim on viability. Does the peptide violate the cell wall but not the cell membrane?

  • You have two Figure 10’s, the eukaryotic cells should be Figure 11. Also, you mention using standard error in Figure 11. Is that appropriate here rather than standard deviation? If so, why? Also, how many replicates are averaged here? And what do you mean by with and without incubation? Please update the figure legend.

  • Line 269: You should run a statistical test to be able to make the claim that viability was not “significantly affected”. What is significant? Both of your graphs show a downward trend with the increase in concentration, especially in B for the breast tumor cells, where about 20% of the cells died (didn’t survive). These are preliminary studies, but running these tests with primary rather than immortilized cells would also be more convincing.

  • Line 271: If you plan to market the drug for use with strains that require 375 uM to see an effect, you should also test that concentration against eukaryotic cells. A statement on why 40 uM is the highest you tested would help here.

  • Line 279: What is “significant antimicrobial activity”? A statistical test for the significance of your results throughout the study would be useful. Although you see convincing effects in many cases, it’s a bit of a stretch to say the peptides inhibit microbes at the same level as gentamicin sulfate. There are differences in the concentrations used for the peptides vs antibiotic that might explain this, but the reasoning for the concentrations chosen is never explained.

  • Line 316: What are the strain-specific resistance profiles you reference here? The differences in the antimicrobial effects against the two MRSA strains used in your study was quite notable and one of the more interesting aspects of the study. You should expand on what is different about the strains and draw more conclusions about why the peptide worked better for one strain vs the other, as appropriate, rather than requiring the reader to visit your reference. It’s notable from your methods section that they require growth in different media. Could that play a role?

  • Since you used the ribosomal S1 protein of S. aureus to create your peptides, are you surprised that they do not perform better against S. aureus? This would be an interesting point to discuss.

Minor Corrections:

  • Page 1 line 46: Run-on sentence difficult to interpret. Make two sentences or use a semi-colon if these ideas are connected. Ex. with semicolon: “The advantage of AMPs over traditional antibiotics lies in their multiple spectrum of action, and the rapid antimicrobial action of AMPs in combination with biodegradability; as a rule, low toxicity can be supplemented by the use of composites, various expression, and delivery systems [6–8].”

  • Please note # of replicates and statistical methods used to produce the graphs and error bars in all of the figure legends, not just in the methods.

  • The peptide naming scheme and abbreviations can be hard to follow because they are so similar to each other (R23XX). Consider making a table and calling them peptides 1 through 3, or just referring to the endings of the current peptide names: peptides F, DI, and EI, or something similar. You could also refer to them by the domains they are contained in after you down-select and define everything. Even a dash for emphasis in the differences within the very similar names as they currently are would be quite helpful and is the easiest fix: R23-F, R23-DI, and R23-EI.

Author Response

Please see the attchment.

Reviewer 2 Report

The mnuscript entitled "Multiple Antimicrobial Effects of Hybrid Peptides Synthesized Based on the Sequence Ribosomal S1 Protein from S. aureus" reporte hybrid antimicrobial peptides containing amyloidogenic regions of the ribosomal S1 protein from Staphylococcus aureus. The hybrid peptides R23F, R23DI, and R23EI can be used as antimicrobial peptides against gram-positive and gram-negative bacteria. However, there are several considerations that need to be reviewed as follow:

  1. The content needs to be refined.
  2. S. aureus should be given its full name in title.
  3. Data on antimicrobial activities of peptides V10F, G10G, V10I, Q10I and L10I need to be supplemented to contrast the activities of hybrid peptides.
  4. Explain the basis for selecting RKKRRQRRRGGSarG at the N-terminal of peptides and its effect on the antimicrobial activity of hybrid peptides.
  5. What is the international level of antimicrobial activity of the hybrid peptides reported in this paper? Is there any relevant report? It is suggested to add the comparison of antimicrobial activity with other reported antimicrobial peptides in the discussion.
  6. It is suggested to improve the antimicrobial mechanism of antimicrobial peptides, such as cell wall permeabilization and/or cytoplasmic membrane depolarization.
  7. MIC of antimicrobial peptides should be supplemented.
  8. line 231, which figure?
  9. In the discussion, it is suggested to discuss relevant antibacterial experiments data of AMPs or hybrid AMPs that have been reported, so as to facilitate understanding of the effects of the hybrid peptides described in this paper and increase persuasibility.
  10. Figure 7B R23F R23DI   There is a large error bar of some data in the figure. Do you need further verification?
  11. The HPLC and CD circular dichroism spectrum profiles of the synthetic peptides should be supplemented.

Author Response

Please see the attchment.

Round 2

Reviewer 1 Report

The authors provided thoughtful responses to my prior comments and provided clarity both in those responses and within changes to the manuscript. The results and discussion have significantly improved.

In the future, please refer to corresponding line numbers in your paper when noting changes made to a manuscript to make finding your improvements related to specific comments easier.

Comments:

  1. Error in title, needs “of”: “Multiple Antimicrobial Effects of Hybrid Peptides Synthesized Based on the Sequence (of) Ribosomal S1 Protein from Staphylococcus aureus”

  1. Thank you for supplementary table S4. However, further explanation is needed in the text as to why V10F and V10I were selected based on those results, as it is still unclear how you’ve interpreted those results to come to this conclusion. It seems to potentially be related to the results with the MRSA (SA 180-F) strain specifically, since that is what those two peptides have in common? Please explain.

  1. With regard to section 2.6 and Figure 12, the wording of “did not significantly suppress the viability” remains without proof that a ~20% increase in cell death is insignificant for the eukaryotic cell lines used for these toxicity tests. Please reword or provide statistical tests towards the claim that ~20% cell death is not significant. Whether “significant” or not, it does look promising. But Table 1 showing percentage of cells in each of the growth phases does not explain why so many cells died, even if you use it to suggest the cells that survived are viable and comparable to controls.

  1. The conclusion should follow the discussion rather than the methods. It’s a bit of an awkward placement in the manuscript.
